# Cultivar-specific nutritional status of potato (*Solanum tuberosum* L.) crops

**Zonlehoua Coulibali[1], Athyna Nancy Cambouris[2], Serge-Étienne Parent [1]***

**1** Department of Soils and Agrifood Engineering, Université Laval, Québec City, Québec, Canada, **2** Quebec Research and Development Centre, Agriculture and Agri-Food Canada, Québec City, Québec, Canada

\* serge-etienne.parent.1@ulaval.ca

**Data Availability Statement:** All relevant data are within the paper and its Supporting Information files. There is no restriction on sharing of data and/or materials.

## Abstract

Gradients in the elemental composition of a potato leaf tissue (*i.e.* its ionome) can be linked to crop potential. Because the ionome is a function of genetics and environmental conditions, practitioners aim at fine-tuning fertilization to obtain an optimal ionome based on the needs of potato cultivars. Our objective was to assess the validity of cultivar grouping and predict potato tuber yields using foliar ionomes. The dataset comprised 3382 observations in Québec (Canada) from 1970 to 2017. The first mature leaves from top were sampled at the beginning of flowering for total N, P, K, Ca, and Mg analysis. We preprocessed nutrient concentrations (ionomes) by centering each nutrient to the geometric mean of all nutrients and to a filling value, a transformation known as row-centered log ratios (clr). A density-based clustering algorithm (*dbscan*) on these preprocessed ionomes failed to delineate groups of high-yield cultivars. We also used the preprocessed ionomes to assess their effects on tuber yield classes (high- and low-yields) on a cultivar basis using k-nearest neighbors, random forest and support vector machines classification algorithms. Our machine learning models returned an average accuracy of 70%, a fair diagnostic potential to detect in-season nutrient imbalance of potato cultivars using clr variables considering potential confounding factors. Optimal ionomic regions of new cultivars could be assigned to the one of the closest documented cultivar.

## 1 Introduction

Potato cultivars are commonly classified into maturity groups based on the number of days from planting to maturity [1]. Compared to other maturity groups, cultivars with longer maturity generally show yield potential that is similar or higher [2–4] because of higher genetic potential [5] related to higher foliar nitrogen status [6] and root acquisition rate [7]. Hence, nutrient management of potato cultivars often consider the cultivar maturity group. However, nutrient profiles or ionomes [8, 9] may vary among potato cultivars of the same maturity groups because cultivars inherit from a diversity of parents specific traits for nutrient absorption and assimilation [10]. Indeed, White et al. [11] provided evidence of important ionome variations in angiosperm species and stated that plant families could be distinguished by their shoot ionomes. Successful classifications of plant species based on axis-reductions have been

**Funding:** ZC is partly funded by the Natural Sciences and Engineering Council of Canada (CRDPJ 385199-09 and DG-2254 - https://www.nserc-crsng.gc.ca), the Quebec Ministry of Agriculture, Fisheries and Food (IA216581 - https://www.mapaq.gouv.qc.ca), Centre SEVE (https://centreseve.recherche.usherbrooke.ca/), Patate Dolbec Inc. (https://patatesdolbec.com/), Groupe Gosselin FG (http://gosseling2.com), Agriparmentier Inc., Ferme Daniel Bolduc Inc. (http://fermedanielbolduc.com/), Patate Laurentienne, Ferme Bergeron-Niquet, and Patates Lac-St-Jean (http://plsj.ca/). There was no additional external funding received for this study. The funders had no role in study design, data collection and analysis, decision to publish, or preparation of the manuscript

**Competing interests:** All the funders (Natural Sciences and Engineering Council of Canada, Quebec Ministry of Agriculture, Fisheries and Food, Centre SEVE, Patate Dolbec Inc., Groupe Gosselin FG, Agriparmentier Inc., Patate Laurentienne, Ferme Bergeron-Niquet, and Patates Lac-St- Jean) have declared that no competing interests exist. This does not alter our adherence to PLOS ONE policies on sharing data and materials.

implemented on compositionally preprocessed plant ionomes [12, 13]. The potato cultivar may also be classified similarly, allowing newly introduced cultivars to benefit from the documented nutrient management of older cultivars. Hence, the foliar ionome, easily collected from field trials, could provide a tool for the fertilization of newly introduced cultivars.

Tissue ionome portrays plant nutritional status [13] under the assumption of causal relationships between plant growth rate and nutrient concentration in a tissue [14, 15]. In survey datasets, reference compositions are those that are nutritionally balanced [12]. Imbalanced ionomes could be rebalanced theoretically through a perturbation operation [16] *i.e.*, a change in tissue composition after nutrient stress has been applied. Any factor impacting yield response to nutrients can perturb leaf composition [17]. Fertilization perturbs soil composition [18] by supplying readily available plant-nutrients [19].

Because nutrients interact in the plant, Baxter [20] suggested that the ionome could be treated as a combination of elements rather than elements taken in isolation. Parent [13] described ionomes as multivariate balance systems of isometric log-ratios [16]. Isometric log-ratios maps vectors of concentrations, which are strictly positive data constrained to the measurement unit that convey only relative information, to a real space of orthonormal coordinates [21]. Indeed, ionomes are intrinsically multivariate: each part cannot be interpreted without being related to the other parts of the whole [22]. Parent and Dafir [23] developed the compositional nutrient diagnosis in plants using row-centered log-ratios (clr). Thereafter, compositional data transformation has been used to preprocess combined nutrients traits of plant species and cultivars [13, 24–26] as well as animal species [27], and human food [28, 29].

The first objective of this study was to identify clusters of potato cultivars based on their leaf ionomes. The second objective was to develop, evaluate and compare the performance of machine learning algorithms in predicting yield categories using ionomes. The third objective was to develop a conceptual workflow to adjust the ionome of potato cultivars using compositional perturbations. Our hypotheses were that (1) nutritionally balanced leaf ionomes of potato cultivars differ among potato cultivars, (2) tuber yield is impacted by specifically leaf compositional traits, and (3) cultivar-specific leaf ionomes could be rebalanced using a perturbation operation.

## 2 Methodology

### 2.1 Data set

The data set is a collection of potato surveys, and nitrogen (N), phosphorus (P) and potassium (K) fertilizer trials conducted in the province of Québec (Canada) from 1970 to 2017 (S1 Table) between the US border (45th parallel) and the Northern limit of cultivation (49th parallel). The data set was filtered to remove foliar samples collected too early or too late from the beginning (10%) of flowering, as reported by scouting teams, and where three or more of the five major elements (N, P, K, Ca and Mg) have not been quantified. The complete data set comprised 3382 observations of 151 field trials. Five maturity classes were represented, and we matched the duration from planting to harvest described by the Canadian Food Inspection Agency [1], although the names differed: early season (65–70 days), early mid-season (70–90 days), mid-season (90–110 days), mid-season late (110–130) and late season (130 days and more) cultivars. The number of samples per cultivar and the corresponding maturity classes are reported in S2 Table.

### 2.2 Diagnostic tissue composition

The potato diagnostic tissue is the first mature leaf (4th leaf from top) sampled at the beginning (10%) of the blooming stage [15, 30]. Twenty to 30 leaves were collected at random in each

plot, composited, dried at 65˚C, ground to pass through a 1 mm sieve, and analyzed for N, P, K, Ca and Mg concentrations after dissolution of combustion. Total N was determined by micro-Kjeldahl or Dumas combustion (Leco CNS-2000 analyzer, St. Joseph, MI, USA). After acid dissolution [31], K, Ca, and Mg concentrations were quantified by atomic absorption spectrometry or inductively coupled plasma spectroscopy (ICP), and P by colorimetry or ICP. We made no distinction between methodologies in the analysis of ionomes.

## 2.3 Processing nutrient composition to nutrient balances

The compositional space [16] of the leaf tissue comprised five nutrients (N, P, K, Mg, Ca) and undetermined components amalgamated into a filling value (Fv) computed by difference between the measurement unit and the sum of quantified nutrients. Tissue components were preprocessed using the row-centered log-ratio transformation, as follows [23]:

$$clr_i = ln\left(\frac{x_i}{g(x)}\right) \tag{1}$$

where $x_i$ is raw concentration of the i[th] component and $g(x)$ is the geometric mean across components including the filling value.

## 2.4 Clustering cultivars

Yield thresholds are useful for decision-making. Because tuber yield potential varies widely among cultivars, we processed by discretizing tuber yields into low- and high-productivity categories [12] by ranking the marketable yield in ascending order within a given cultivar, and selecting the yield corresponding to the 65[th] percentile as cut-off between the two subgroups. Hence, each cultivar had its cut-off with respect to its yield potential as shown in S2 Table. The high-yielding subpopulation ionomes were used to assess cultivars clustering ability. This subgroup comprised 1190 occurrences (after the exclusion of 144 outliers) across 151 trials and 47 cultivars. A density-based clustering method [32] was used to delineate cultivar groups of similar compositions using clr variables.

## 2.5 Ionome effect and yield prediction

Machine learning algorithms can either regress to predict continuous variables or classify to predict categories [33]. Tuber yield categories were predicted using clr variables and information on ionomic groups of the full data set (high and low yielders *i.e.* 3382 rows). Three machine learning algorithms were compared: k-nearest neighbors, random forest and support vector machines.

We estimated the relative influence of variables in the model and their rank by examining how can prediction error increases when data for a variable is permuted while all others are left unchanged [34, 35]. A variable can score a zero or too small value compared to others. Deleting such variable from the dataset should not impact on the results. The random forest model was used for feature selection to assess importance of each clr variable in predicting tuber yield, but none of the variable was removed.

The data were split into training (75%) and testing (remaining 25%) sets at cultivar level i.e., for each cultivar the samples were randomly separated according to these proportions. The performance of the classification models was assessed using accuracy computed with the testing set. Applied to the context, the four quadrants defined by Swets [36] in binary system diagnosis to delineate the response classes are presented in the contingency table (Table 1).

**Table 1. Term definitions used for the study.**

| | | Observed yield | |
|---|---|---|---|
| | | Low (unbalanced) | High (balanced) |
| **Predicted yield** | Low | **True positive (TP)**: observed low-yielders correctly predicted as low-yielders. | **False positive (FP)**: observed high-yielders incorrectly predicted as low-yielders. |
| | High | **False negative (FN)**: observed low-yielders incorrectly predicted as high-yielders. | **True negative (TN)**: observed high-yielders correctly predicted as high-yielders. |

As in medical sciences, the *negative* term is used in cases where no intervention is needed after diagnosis.

The accuracy is the proportion of correctly-predicted instances:

$$Accuracy = \frac{TN + TP}{TN + TP + FN + FP} \qquad (2)$$

## 2.6 Rebalancing a composition: The enchanting islands

A compositional perturbation is a translation in the compositional space [37, 38]. A perturbation vector expressed as clr values contains a series of deltas (differences). Once back-transformed into the compositional space, the perturbation vector alters a composition through a perturbation ($\oplus$) operation as follows [37]:

$$A \oplus B = [a_1, a_2, \ldots, a_D] \oplus [b_1, b_2, \ldots, b_D] = \mathcal{C}(a_1 \times b_1, a_2 \times b_2, \ldots, a_D \times b_D) \qquad (3)$$

where a D-part composition A is perturbed ($\oplus$) by a D-part composition B, and $\mathcal{C}$ is the closure operator to constant sum.

We used the testing set to display the effect of a perturbation across the simplex. We selected two elements (N and P) and simulated an increase of their initial (observed) clr values by 20% (theoretically). The observed (ionome of the instance) and new clr vector (perturbed ionome) were back-transformed into N, P, K, Ca, Mg and Fv compositional space for comparison using familiar concentration units.

The high yielders of the training set correctly diagnosed as balanced (true negative specimens) by the most accurate model were used as the reference subpopulations. The clr values of these reference specimens were used as reference nutritional status at high yield potential. A potato nutrient imbalance index was computed as a distance from the closest high-yielding specimen using the Aitchison distance, *i.e.* the Euclidean distance between compositions using clr-transformed concentrations [39]. For any misbalanced or new specimen of a given cultivar, the closest true negative (closest reference composition) was identified as the sample with the minimum Aitchison distance from the new composition. The nutrient clr differences defining the Aitchison distance may be considered as apparently excess or deficiency of the nutrient requiring correcting measures in a multivariate and compositional data perspective [40]. Hence, the clr space of nutrient components (N, P, K, Ca, Mg) was described not as an ellipsoidal hyper-space [41] but as islands of high-yielding specimens dispersed in the hyper-space of differently yielding specimens. The closer is a specimen from the enchanting island, the higher its chance to become a high-yielder [40]. The clr-difference was converted into a perturbation vector between two nutrient compositions expressed as familiar nutrient concentrations.

## 2.7 Statistical analysis

Statistical computations were performed in the R statistical environment version 3.6.1 [42]. Compositional data analysis was conducted using the R *compositions* package version 1.40–2 [43]. Multivariate outliers were removed for robust multivariate analysis [44] using the

Mahalanobis distance at a 0.01 level of significance with the R *mvoutlier* package version 2.0.9 [45]. The clustering operation were performed using *dbscan* package version 1.1–3 [32]. Linear discriminant analysis (LDA) was conducted using the R *ade4* package version 1.7–13 [46] which allows computing linear combinations of clr coordinates that best discriminate cultivars ionomes centroids. Supervised analysis was conducted using the *caret* package version 6.0–84 [47]. Our results are reproducible by using the R computation codes and data given as supplementary information and available online in a GitHub repository (https://git.io/Jvt2r).

## 3 Results

### 3.1 Cluster analysis

The data set used for clustering is described in S2 Table. The AC Chaleur cultivar showed the lowest tuber marketable yield cut-off (65th percentile) at 17.4 Mg ha$^{-1}$ and Red-Maria, the highest at 64.6 Mg ha$^{-1}$. Average marketable yield was 40.5 Mg ha$^{-1}$ for high yielders and 24.8 Mg ha$^{-1}$ for low yielders. In comparison, average potato tuber yields in Canada and Québec were 31.2 Mg ha$^{-1}$ and 32.2 Mg ha$^{-1}$ respectively, in 2018 [48].

The *dbscan* clustering function looked for dense regions in the clr-space, and detected a single cluster of cultivars *i.e.*, cultivars were scattered without any particular dense region. A principle components analysis allowed to map cultivars and nutrients in the biplot shown in Fig 1. The principle components scores mapped on the distance biplot (Fig 1A) showed no particular pattern allowing groups partition. The clr correlation loadings (Fig 1B) showed a negative

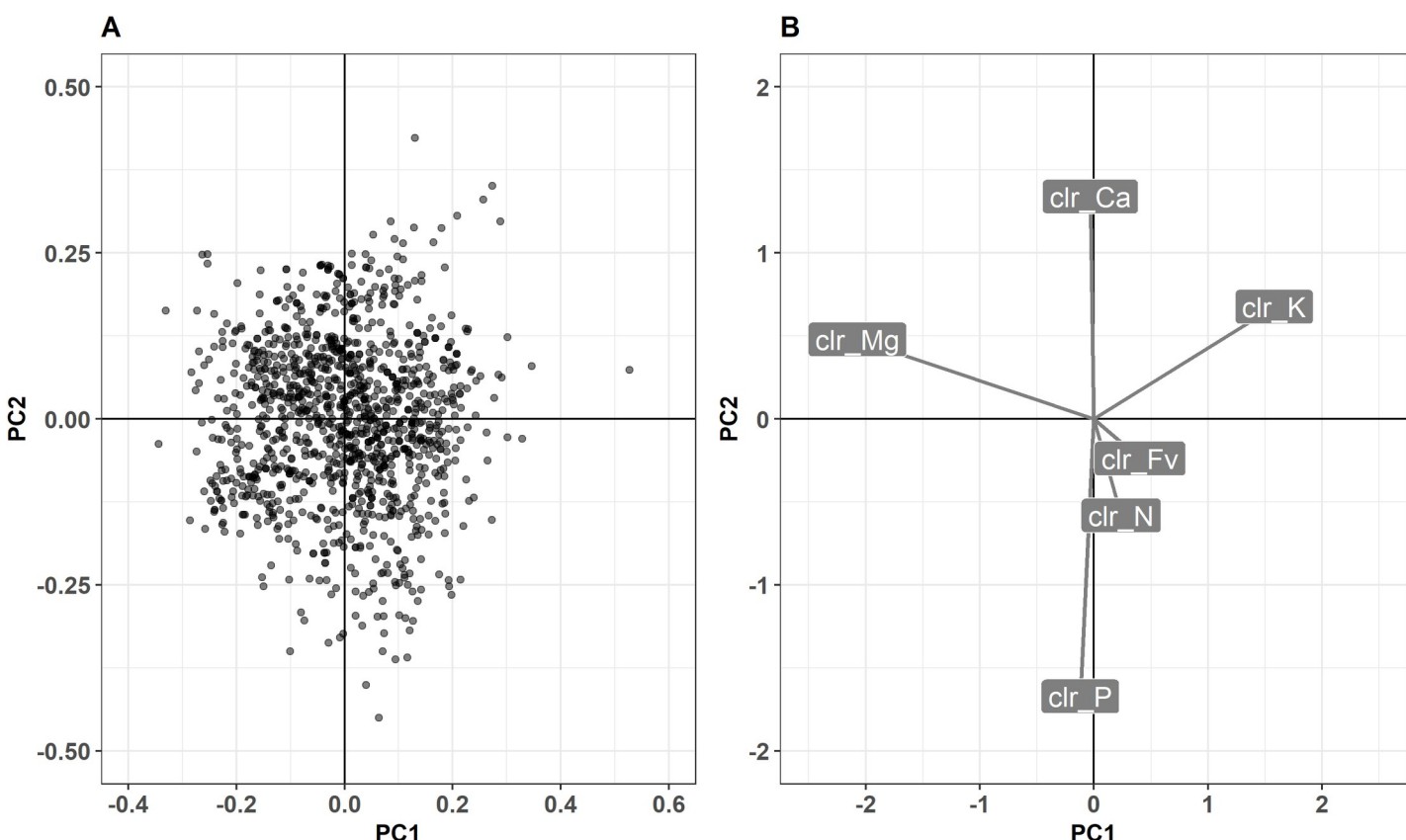

**Fig 1.** Principle components biplot of potato ionome showing (A) scores in distance scaling and (B) loadings in correlation scaling.

relationship between K and Mg, P and Ca, and positive relationship between N and P in agreement to concentration changes with time as the plant matures [49]. Discrepancies between cultivars were driven mainly by Mg and K on the first axis, and by P and Ca on the second axis (right hand side plot).

## 3.2 Predicting tuber yield

Classification models assigned explanatory clr variables to two categorical tuber marketable yield: high- and low-yielders. The random-forest algorithm allowed to rank the importance of variables in the model. The clr of nitrogen appeared to be the most discriminant variable between tuber yield categories, followed by the amalgamated unknown components (Fv), then Ca, Mg and, finally, P.

After splitting data into training (75%) and testing (25%) data sets, we used a ten-fold cross-validation process that sequentially splits the training data set into ten parts, using nine parts for calibration and the remainder for validation. The $k$-nearest neighbours, the random forest and the support vector machine models returned practically similar predictive accuracies (although slightly lower for the support vector machine algorithm), with a mean accuracy of 70% representing 591 successful and 254 unsuccessful cases classification with the testing set. The null hypothesis for a random classifier *i.e.*, non-informative classification consisting of an equal distribution of 50% successful and 50% unsuccessful cases was rejected after a $\chi^2$ homogeneity test ($\chi^2 = 69.135$, p $< 2.2$ $10^{-16}$). Since all the models returned practically similar accuracy over the testing set, predictions with the $k$-nearest neighbors model were used for interpreting. There was high variation in model fit by cultivar as shown in Fig 2. The accuracy at testing varied from 25% for Estima and Waneta, to 100% for Ambra, Carolina, Dark Red

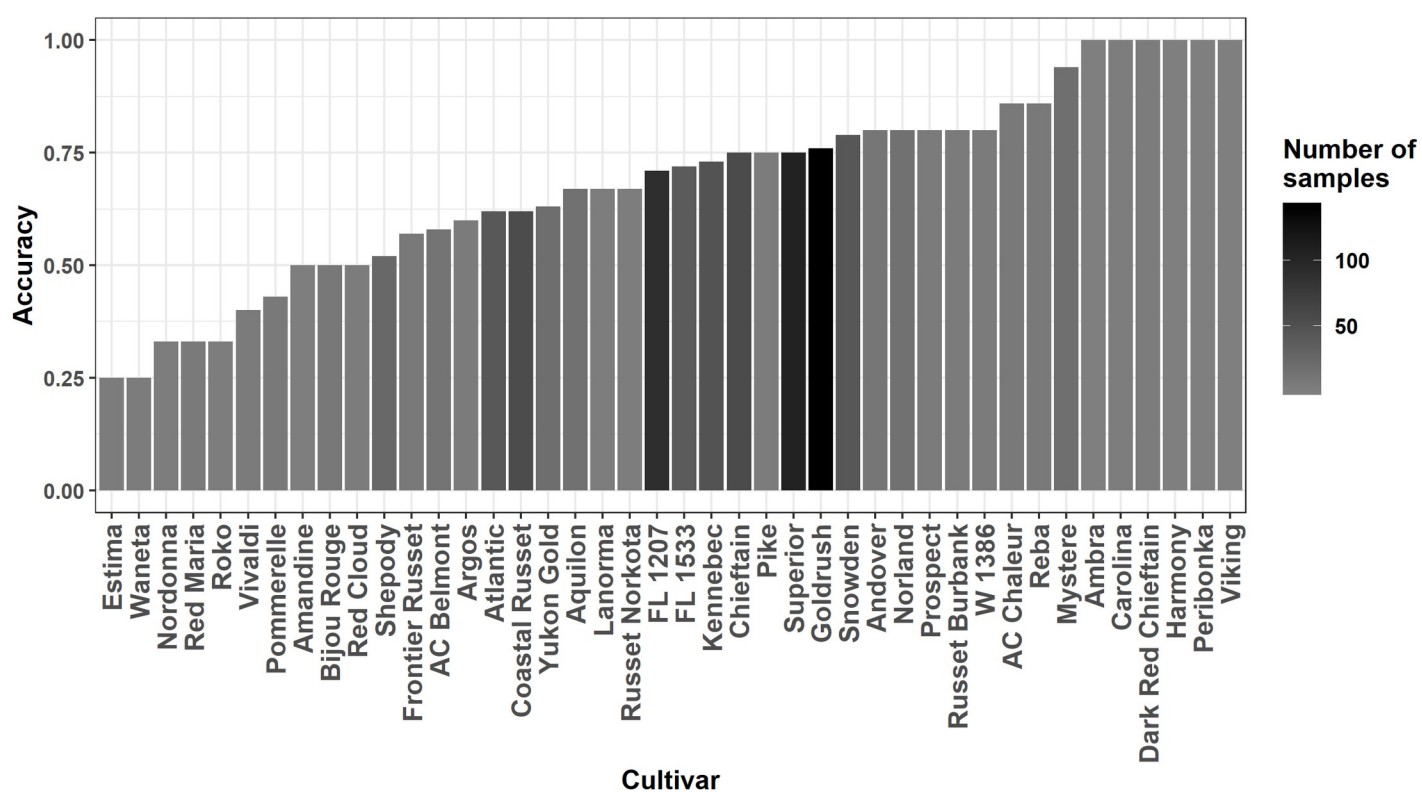

**Fig 2. The k nearest neighbors model evaluation accuracies for cultivars.**

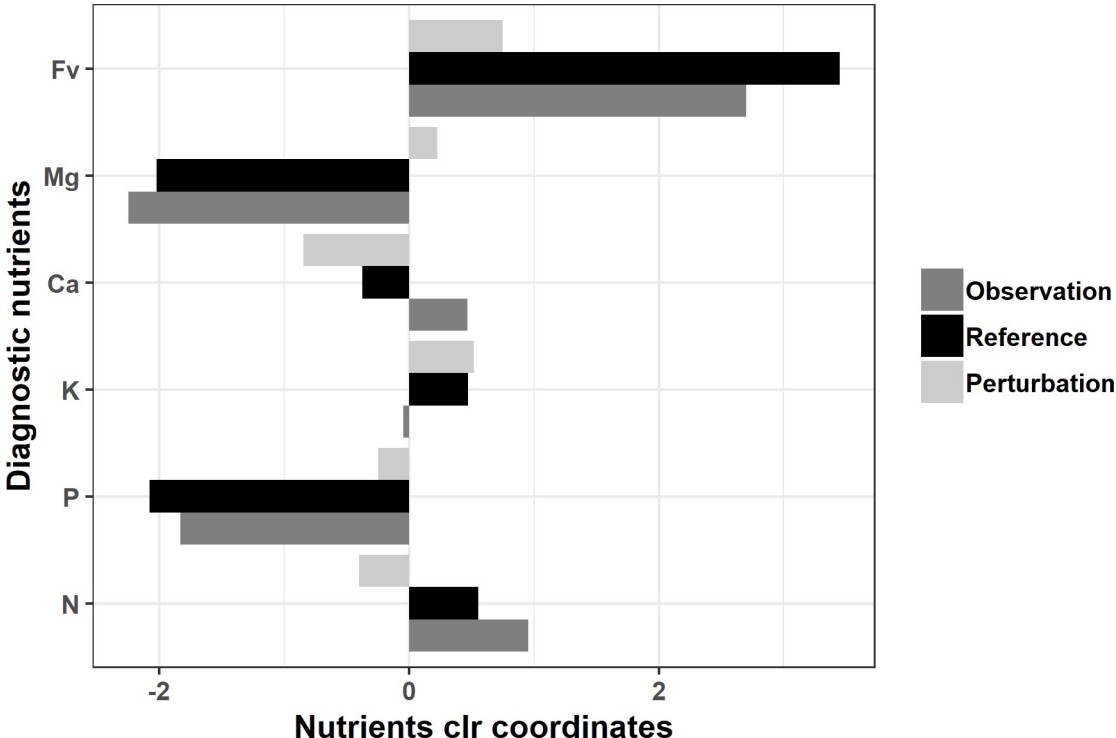

**Fig 3. Perturbation vector example mapped using the most imbalanced sample.** The most imbalanced observation nutrient composition was (0.0601, 0.0037, 0.0355, 0.0032, 0.0048. 0.8919), the nearest reference composition was (0.0561, 0.0036, 0.0603, 0.0052, 0.0184, 0.8565), the corresponding perturbation vector was (0.0919, 0.0965, 0.1696, 0.1629, 0.3832, 0.0959) for N, P, K, Mg, Ca and Fv respectively. The Aitchison distance computed between the observation and its associated true negative was 1.135.

Chieftain, Harmony, Peribonka and Viking. All these cultivars had small sample sizes in the dataset, as shown in the S2 Table.

### 3.3 Ionome perturbation

The true negative specimens (correctly diagnosed as balanced) comprising 783 occurrences in the training data set provided the clr reference values required to compute the Aitchison distance, which is equal to the Euclidean distance across clr-transformed compositions. The S3 Table displays mean values for each cultivar. Using the Aitchison metric, the closest true negative specimen was set as the reference composition for each imbalanced specimen. In the clr-space, the difference between the reference and the imbalanced compositions returns a perturbation vector. The Fig 3 shows the imbalanced sample with the highest Aitchison distance from its reference and the perturbation to apply as a translation to reach a balanced ionome.

## 4 Discussion

### 4.1 Clustering potato cultivars

The Canadian Food Inspection Agency classified potato cultivars broadly into maturity groups based on the time elapsed between planting and maturity [1]. However, nutrient requirements, especially nitrogen, vary widely between cultivars of the same maturity group. In New Brunswick (Canada), Zebarth et al. [50] recommended 200–208 kg N ha$^{-1}$ for Russet Norkota (early-season cultivar) and Russet Burbank (late-season cultivar), 190 kg N ha$^{-1}$ for Superior (early-mid-season cultivar) and Goldrush (mid-season cultivar), 175 kg N ha$^{-1}$ for Shepody (mid-

season), 135 kg N ha$^{-1}$ for early cultivars for the table market, 160–180 kg N ha$^{-1}$ for other mid-season, 180–200 kg N ha$^{-1}$ for other late, and 135–160 kg N ha$^{-1}$ for low N requirement cultivars. Such large discrepancies within the same cultivar maturity group was attributed to differential foliar gene expression [6] and root development [7]. Hence, information additional to maturity grouping is needed to assess nutrient requirements of potato cultivars. Huang and Salt [51] reported that ionomics allows the discovery of genes controlling natural variation in the plant ionome and for Salt et al. [9], ionomics could capture information about the functional state of an organism driven by genetic and environmental factors. The content of plant tissue reflects what the plant can absorb from the soil and for each nutrient, there is a correlation between its concentration and yield. Moreover, since tissue analysis is also carried out to observe the effect of fertilizer applications, and for determining the in-season or next season nutrient requirement [52, 53], ionomes could be useful in discriminating potato cultivars. Indeed, using a small data set of eight potato cultivars, Hernandes et al. [10] showed that foliar nutrient profiles varied widely among cultivars of the same maturity group. According to Parent et al. [12], variations in ionomes could be interpreted only partly as genotypic effect, and phenotypic plasticity can also be driven by nutrient supply capacity specific to agroecosystems while breeding programs are conducted under relatively luxurious environments to reach high productivity. The N, Mg and K clr values, that dominated principal components (Fig 1), could reflect the abilities of individual cultivars to acquire and use those nutrients more efficiently [54, 55]. Natale et al. [56] provided evidence that in general macronutrient contents differ among species and cultivars and within the same species for fruit trees. For N, K and Ca, this range is wider because of higher requirement of these elements by plants, and narrower for P, Mg and S, indicating smaller demand for the latter.

To cluster is to recognize that objects are sufficiently similar to be put in the same group, and to identify distinctions or separations between groups of objects [57, 58]. Based on the assumptions of differential genotypic potential, root development, nutrient requirements, nutrient uptake and use efficiency, the goal was to discover interesting structures in the N, P, K, Mg and Ca contents of the diagnosis tissue in order to decipher dissimilarities between cultivars [33]. However, the process failed to discriminate groups of cultivars along the clr coordinates. Hernandes et al. [10] reached similar results with overlapping nutrient profiles between cultivar groups depending on isometric log ratio (ilr) coordinates. They found similar nutrient profiles between cultivars groups along some ilr coordinates and very different ones along others. While ionome dissimilarities are not numerically compelling, they could assist classifying new cultivars into appropriate ionomic group to benefit from costly fertilizer trials conducted on cultivars of the same group.

## 4.2 Tuber marketable yield prediction

The P content of the diagnostic leaf did not appear useful in predicting potato tuber yield classes. Other elements (N, K, Ca and Mg) showed important contribution of their clr values to the prediction quality metric, especially N, which is directly related to photosynthesis [59]. Since the fertilization trials were conducted over a time span of 47 years (1970–2017), the question arises whether the different methodology of quantifying P (colorimetry/ICP) may have contributed to depreciating this variable in predicting tuber yield classes. The ICP method is shown to be faster and to give higher results for total phosphorus content in 'soil' extracts in comparison to the colorimetric method. However, there are exceptions and controversial results [60–62]. Ivanov et al. [61] found that the two methods for total P determination in plant material were highly correlated, and the results were generally within 5% to 10% of one another. Moreover, Valkama et al. [63] reported that, although agricultural practices, soil

conditions and analytical techniques have undergone substantial changes over time, the differences between old and recent experiments in yield responses to P application were not statistically important. For all these reasons, we consider the two analytical methods equally relevant to the analysis. The low importance of the P clr variable in predicting tuber yield classes may come from its correlation with Ca. Globally, the selection of relevant features is achieved, by first checking the correlation between features and response to select the features that have correlation above a selected level (*e.g.*, 0.5). Then, the independent variables need to be uncorrelated with one another. If some features are correlated, only one is kept. The process selected the *clr_Ca* variable (alphabetical order) instead of *clr_P* since these features are correlated as shown in Fig 1B. In this study no element was discarded from the process relative to its importance.

The tested algorithms (*k*-nearest neighbours, random forest and support vector machine) returned similar accuracies in the prediction of yield classes using clr variables as predictors and showed fair diagnostic potential to detect nutrient imbalance. The correctly predicted high and low yielders reached 70% in the testing data set. The models classified more accurately the yield categories compared to a random classifier [64]. Specimens classified as false negatives (*i.e.*, low yielders incorrectly classified as high yielders) are attributable to limiting conditions other than N, P, K, Mg, and Ca nutrition: soil physical and chemical properties [65, 66], fertilization [67], management failures, diseases [68] or weather events [69] impacting plants growth and yield potential. False positive specimens (*i.e.*, high yielders incorrectly classified as low yielders) indicate luxury consumption when nutrient concentrations are higher [12, 70], or other particularly favorable growth conditions. The confusion matrix built for cultivars revealed poor predictive accuracy for certain cultivars (*i.e.*, 25% for Estima and Waneta) and conversely an accuracy of 100% for others (*i.e.*, Ambra, Peribonka) as shown in Fig 2. These cultivars involved mainly small sample sizes (only one, two or three high-yielders and five, six or lightly more low-yielders). The problems of small-data in machine learning are numerous, but mainly revolve around over-fitting. The training and testing datasets division could only aggregate observations of one class in the training set so that the model would train to always predict this dominant class [71]. The model could also memorize labels, which is not ideal for generalizing from new data. Brownlee [72] explained that imbalanced classifications (one or less examples in a minority class for hundreds or more examples in the other) pose a challenge for predictive modeling as most of the machine learning algorithms used for classification were designed around the assumption of an equal number of examples for each class. This results in models that have poor predictive performance, specifically for the minority class. The controversial accuracy level for some cultivars (especially low level) could also come from other yield limiting factors specific to the experiments but not involved in this study, as for false positive specimens. Our model was not effective for these cultivars treated separately.

The differential nutrition of potato cultivars could be addressed objectively using mineral analysis of the diagnostic leaf. More data are needed for poorly documented cultivars. Moreover, dedicated models could be trained for cultivars for which sufficient data are available (e.g., Goldrush, Superior, FL 1207, Chieftain). Other algorithmic, sampling and quality measurement approaches could further be implemented to deal with the problems of small-data and unequal distribution of classes [71, 72]. One could extend the predictors to the experimental conditions (soils, weather data), fitting a site-and-cultivar-specific nutrients diagnosis model.

## 4.3 Perturbation vector for fertilizer recommendation

Rational fertilization requires information on the nutrients that are available in the soil, and the nutritional status of the plant [14] as portrayed by the diagnostic tissue composition [14,

15]. However, the diagnosis of deficiency and toxicity of mineral nutrients may be complicated in field-grown plants where more than one mineral nutrient is deficient or where there is a deficiency of one nutrient and simultaneously toxicity of another [14]. The scientific principle behind tissue analysis is that healthy plants contain predictable concentrations of analytical nutrients [73]. The values are compared to established norms for inadequate, adequate and excess levels. However, Parent et al. [13] proved that this concept of growth-limiting nutrient concentrations supported by the *Law of minimum* and illustrated by Liebig's barrel, should be replaced by a concept of growth-limiting nutrient balances illustrated by a pan balance design, where groups of elements are balanced optimally against each other in weighing pans.

The difference between two equal-length compositional vectors can only be computed using tools of compositional data analysis. The perturbation vector concept applied to foliar tissue diagnosis returns a scaling operator [21] that when applied to an imbalanced composition translates it (theoretically) into a balanced composition with high yield potential (*i.e.*, true negative). Although the closure of the simplex implies that a perturbation on the clr of a specific nutrient is methodologically not a change in proportion of a single nutrient, perturbations expressed in the clr space appear suitable for interpretation. Indeed, the difference measured between clr values of the diagnosed sample and reference (true negative) specimen can be ranked using the sign of that difference [10, 74, 75], hence indicating which components are at excessive or deficient levels. As provided by Parent [40], K and Mg were apparently deficient while N, P and Ca were apparently in excess compared to the closest *reference* specimen (Fig 3). Using the same approach, ionomes of newly introduced cultivars with unknown nutrient requirements could be assigned to the cultivars of known nutrient requirements showing the closest ionomes.

A perturbation as the one shown in Fig 3 should not be interpreted as shifts of individual components, since the operation on a single component resonates on the whole simplex [40]. For instance, an offset in the simplex $S$ (N, P, K, Ca, Mg, Fv) composition following the increase by 20% (theoretically) of N and P clr values is displayed on Fig 4. The K, Ca and Mg concentrations seemed more stable with respect to the others. Although P clr values have been increased, P proportion decreased globally for the new equilibrium of the simplex. The offset was higher for the selected components followed by the filling value (Fv).

Perturbation (as defined in Eq 3) is the measure of compositional change from one composition to another [37]. Because foliar composition belongs to compositional data family, the Fig 4 illustrates the principle that changing a proportion of such data affects at least another proportion of the simplex [16]. The result displayed variable offsets for other elements, decreasing or increasing to reach another balance in the simplex.

## 5 Conclusion

Since the concept of compositional data analysis was applied to plant tissues, several studies classified plant species and cultivars using multivariate analysis of nutrients compositions. This study is, to our knowledge, the third (following Parent et al. [49] and Hernandes et al. [10]) to use statistical tools to address the differential nutrition of potato cultivars using combination of nutrient concentrations in the diagnostic leaf, and the first using tools of machine learning to predict tuber marketable yield. The potato ionomes showed some dissimilarities in principle components analysis, but not compelling to separate definite density-based clusters between cultivars on the basis of the clr values. However, the ionome showed a determinant effect on tubers yield. Used as predictors in machine learning tools, clr variables showed diagnostic potential to detect in-season nutrient imbalance to address objectively the differential response of cultivars to fertilization. The perturbation vector of the leaf compositional space

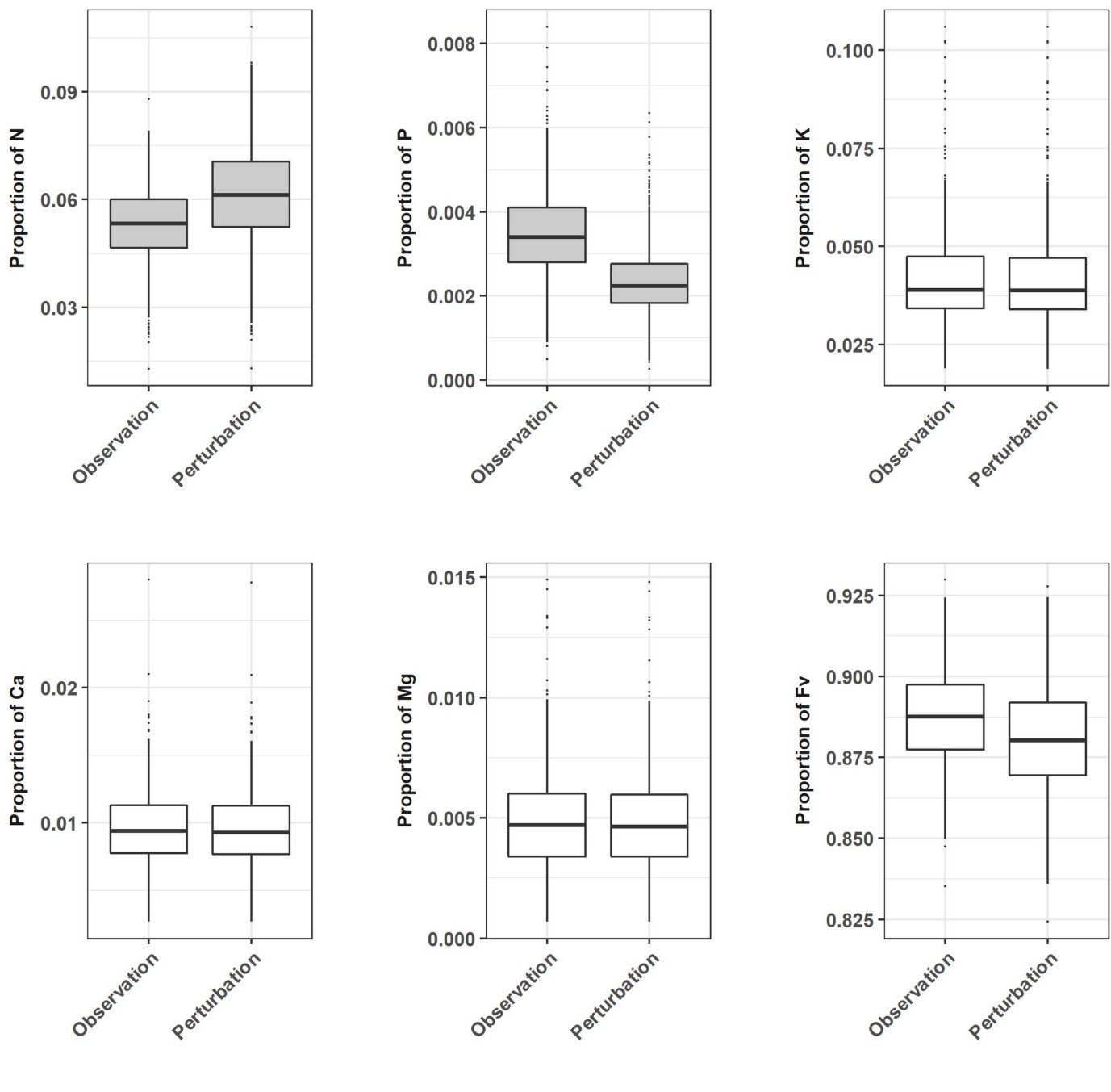

**Fig 4. Effect of the perturbation of N and P clr coordinates on the other element proportions.** 'Observation' stands for the element's original proportion, 'Perturbation' designates the new proportion after the 'Observed' vector's clr value was offset. Greyed boxplots plot distribution of perturbed elements of the simplex.

could indicate cultivar sensitivity to fertilization and address specific problems of nutrient imbalance in new cultivars. Tissue testing remains an informative, diagnostic and preventive tool with real-world applications for growers in evaluating the effectiveness of their nutrient

management program. When using the right interpretation, this timely and correct tissue testing helps diagnose the presence and magnitude of suspected nutrient deficiencies. By using the compositional perturbation vector involving interactions among nutrients, our study provided a useful tool in potato precision fertilization in Quebec. The perturbation vector can help identify limiting nutrients requiring correcting measures as a season progresses or for subsequent seasons. Moreover, our study implicitly provided robust multi-nutrient norms for potato crops, gathering more cultivars of different maturity classes than the previous works. These norms are sets of true-negative or nutritionally-balanced compositions per cultivar (enchanting islands) with high-yield potential. More data are needed to fine-tune the models, especially for poorly-documented cultivars. New algorithms, other sampling methods and model quality measures could be tested to deal with the problem of small-data and imbalanced classification. Further studies extending predictive features to site-specific conditions could improve the diagnosis with a site- and cultivar-specific nutrient diagnosis model.

## Supporting information

**S1 Table. Quebec potato leaves ionome data set.** raw_leaf_df.csv file available online in data repository at https://git.io/Jvt2r.
(CSV)

**S2 Table. Potato data set used for cluster analysis.**
(DOCX)

**S3 Table. True negatives mean clr values for cultivars.**
(DOCX)

## Author Contributions

**Conceptualization:** Serge-Étienne Parent.

**Data curation:** Zonlehoua Coulibali, Serge-Étienne Parent.

**Formal analysis:** Zonlehoua Coulibali, Serge-Étienne Parent.

**Investigation:** Zonlehoua Coulibali, Serge-Étienne Parent.

**Methodology:** Zonlehoua Coulibali, Serge-Étienne Parent.

**Project administration:** Serge-Étienne Parent.

**Software:** Zonlehoua Coulibali, Serge-Étienne Parent.

**Supervision:** Athyna Nancy Cambouris, Serge-Étienne Parent.

**Validation:** Zonlehoua Coulibali, Serge-Étienne Parent.

**Visualization:** Zonlehoua Coulibali, Serge-Étienne Parent.

**Writing – original draft:** Zonlehoua Coulibali, Serge-Étienne Parent.

**Writing – review & editing:** Zonlehoua Coulibali, Athyna Nancy Cambouris, Serge-Étienne Parent.

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
