## [Decision Letter · Decision Letter 0]

27 Dec 2019

PONE-D-19-27444

Cultivar-specific nutritional status of potato (Solanum tuberosum L.) crops

PLOS ONE

Dear Dr. Parent,

Thank you for submitting your manuscript to PLOS ONE. After careful consideration, we feel that it has merit but does not fully meet PLOS ONE’s publication criteria as it currently stands. Therefore, we invite you to submit a revised version of the manuscript that addresses the points raised during the review process.

We would appreciate receiving your revised manuscript by Feb 10 2020 11:59PM. To enhance the reproducibility of your results, we recommend that if applicable you deposit your laboratory protocols in protocols.io, where a protocol can be assigned its own identifier (DOI) such that it can be cited independently in the future. For instructions see: http://journals.plos.org/plosone/s/submission-guidelines#loc-laboratory-protocols

We look forward to receiving your revised manuscript.

Kind regards,

Paul Esker

Academic Editor

PLOS ONE

Additional Editor Comments:

This paper presents a machine learning approach to modeling tuber yields as a function of foliar ionomes. The concept is interesting and the database extensive considering time and cultivar diversity. Overall, the paper does add something new to the literature, but as indicated by the primary reviewer, revisions are required before the article is ready for publication. I would like to apologize for the delay in returning this review since there were challenges with finding reviewers, as well as having several situations where the reviewer was released from finalizing the review due to non-response. Nonetheless, both the primary reviewer and I are in agreement regarding areas for improvement for this manuscript.

In my case, I was confused somewhat by the description of how trials were selected, and what the yield range was since in the methods, trials that had less than 28 Mg ha-1 were dropped from inclusion, yet the low yielding group (high versus low) averaged 24.8 Mg ha-1. I assume this means that within the selected trials, there were still many low-yielding cultivars, correct?

Also, the sample sizes by cultivar were quite variable and in some cases, it appeared that there were very few of one class or the other when looking at the supplementary material. As indicated by Reviewer 1, I think this is important to provide further details or context since it may partially explain the high variation in fit by cultivar - this is not necessarily addressed well in the discussion.

Some more specific observations include:

Lines 229-231: Confusing statement

Line 269 (and in other statements), the citation style was rather odd, with a double mention of the author.

Lines 300-304: Confused with pre-selection procedure - also ties in to the question that Reviewer 1 had for lines 305-307.

Lines 319-323: Seems like a transition is missing to the connect the different thoughts.

Journal Requirements:

"ZC is partly funded by the Natural Sciences and Engineering Council of Canada (CRDPJ 385199-09 and DG-2254), the Quebec Ministry of Agriculture, Fisheries and Food (IA216581), Centre SEVE, Patate Dolbec Inc., Groupe Gosselin FG, Agriparmentier Inc., Ferme Daniel Bolduc Inc., Patate Laurentienne, Ferme Bergeron-Niquet, and Patates Lac-St-Jean. The funders had no role in study design, data collection and analysis, decision to publish, or preparation of the manuscript."

'The authors have declared that no competing interests exist.'

We note that you received funding from commercial sources: Patate Dolbec Inc., Groupe Gosselin FG, Agriparmentier Inc., Ferme Daniel Bolduc Inc.

4. We note that Figure 1 in your submission contains map images which may be copyrighted.

a.    You may seek permission from the original copyright holder of Figure(s) [#] to publish the content specifically under the CC BY 4.0 license.

Reviewers' comments:

Reviewer's Responses to Questions

**Comments to the Author**

1. Is the manuscript technically sound, and do the data support the conclusions?

Reviewer #1: Partly

2. Has the statistical analysis been performed appropriately and rigorously? 

Reviewer #1: Yes

3. Have the authors made all data underlying the findings in their manuscript fully available?

Reviewer #1: Yes

4. Is the manuscript presented in an intelligible fashion and written in standard English?

Reviewer #1: Yes

5. Review Comments to the Author

Reviewer #1: In lines 305-307, the authors state that the sample size is small and results should be carefully interpreted. It would be misleading to the reader to indicate any relationship or conclusions at this stage until further investigation/research is done to explore such implications from this data.

Line 25-26, “The scarcity of data, in particular for new cultivars, constrains to group cultivars into maturity groups.” Awkward sentence – consider revision.

Line 51 “In particular, foliar gene expression….” Incomplete sentence- revise grammar.

Line 109, Can the maturity classes be further described by their range of days from planting to maturity?

Line 113-114, If I understand the protocol correctly, this leaf sampling was taken at different times throughout the growing season due to the differences in maturity of the potato cultivars?

Line 115-116, “ground to less than 1mm” what does that mean exactly? The plant material was ground to less than 1mm particle diameter? Just curious.

Line 265-266, “…information additional to maturity grouping is needed to assess nutrient requirements of potato cultivars.” Can you provide some additional considerations and why? Are they practical?

Line 278-280, The paragraph starts with describing the variation in cultivars and foliar nutrient profiles and would help if there’s more of a transition in explaining to the reader how this variation could be explained through the clustering process. Sentence 278-280 is a stark transition in thought. Consider the additional of another sentence to guide the reader.

Line 290, could the different methodology of quantifying P (colorimetry/ICP) have contributed to the insignificance of its content in predicting tuber yield classes?

Lines 305-307, Expand on why the predictive accuracy for some cultivars were very high while others were not. Would these potential factors need to be considered in future yield prediction models?

Lines 367-369, could the perturbation vector of leaf compositional space assist with correcting for in-season nutrient imbalances (per cultivar) to improve yield potential?

Conclusion section –

This section needs further expansion and discussion. What are the implications of the study on potato fertility and management? What about future directions and next steps? Does this research provide a positive direction in precision potato production in Canada? Does this mean that cultivar-specific nutrition recommendations may need revision or can be more precise in the future? What are the economic implications of this research for the potato industry? Are there any?

Why use potato? Has there been significant background research already conducted in this species or has there been cultivar-specific fertility variability that has led to the investigations with potato? Has there been economic impact of varied fertility regimes on potato cultivars that more precise fertility recommendations based on this model could address?

Could the dataset have been more robust if collected from other geographical sites?

Would it be possible to do some similar analyses with select potato cultivars grown in very controlled conditions and compare these results also to the analyses here?

6. PLOS authors have the option to publish the peer review history of their article (what does this mean?). If published, this will include your full peer review and any attached files.

Reviewer #1: No

---

## [Author Response · Author response to Decision Letter 0]

11 Feb 2020

The reviewers' comments were thoroughly reviewed in our file Response to Reviewers.docx.

---

## [Editor Report · Decision Letter 1]

2 Mar 2020

Cultivar-specific nutritional status of potato (*Solanum*
*tuberosum* L.) crops

PONE-D-19-27444R1

Dear Dr. Parent,

We are pleased to inform you that your manuscript has been judged scientifically suitable for publication and will be formally accepted for publication once it complies with all outstanding technical requirements.

With kind regards,

Paul Esker

Academic Editor

PLOS ONE

Additional Editor Comments (optional):

Thank you for taking the time and energy to consider the reviewer comments. The manuscript reads very well and I am satisfied that it can be moved along for publication.

Reviewers' comments:

None required.

---

## [Editor Report · Acceptance letter]

3 Mar 2020

PONE-D-19-27444R1 

Cultivar-specific nutritional status of potato (Solanum tuberosum L.) crops 

Dear Dr. Parent:

I am pleased to inform you that your manuscript has been deemed suitable for publication in PLOS ONE. Congratulations! Your manuscript is now with our production department. 

With kind regards,

on behalf of

Dr. Paul Esker 

Academic Editor

PLOS ONE